# Contribution of labor related gene subtype classification on heterogeneity of polycystic ovary syndrome

Jue Zhou[1‡], Zhou Jiang[2‡], Leyi Fu[3], Fan Qu[3], Minchen Dai[3], Ningning Xie[3], Songying Zhang[2]*, Fangfang Wang[3]*

**1** School of Food Science and Biotechnology, Zhejiang Gongshang University, Hangzhou, China,
**2** Department of Obstetrics and Gynecology, Sir Run Run Shaw Hospital, School of Medicine, Zhejiang University, Hangzhou, China, **3** Women's Hospital, School of Medicine, Zhejiang University, Hangzhou, China

‡ JZ and ZJ authors contributed equally to this work and share first authorship.
* Drwangfangfang@zju.edu.cn (FW); zhangsongying@zju.edu.cn (SZ)

## Abstract

### Objective

As one of the most common endocrine disorders in women of reproductive age, polycystic ovary syndrome (PCOS) is highly heterogeneous with varied clinical features and diverse gestational complications among individuals. The patients with PCOS have 2-fold higher risk of preterm labor which is associated with substantial infant morbidity and mortality and great socioeconomic cost. The study was designated to identify molecular subtypes and the related hub genes to facilitate the susceptibility assessment of preterm labor in women with PCOS.

### Methods

Four mRNA datasets (GSE84958, GSE5090, GSE43264 and GSE98421) were obtained from Gene Expression Omnibus database. Twenty-eight candidate genes related to preterm labor or labor were yielded from the researches and our unpublished data. Then, we utilized unsupervised clustering to identify molecular subtypes in PCOS based on the expression of above candidate genes. Key modules were generated with weighted gene co-expression network analysis R package, and their hub genes were generated with Cyto-Hubba. The probable biological function and mechanism were explored through Gene Ontology analysis and Kyoto Encyclopedia of Genes and Genomes pathway analysis. In addition, STRING and Cytoscape software were used to identify the protein-protein interaction (PPI) network, and the molecular complex detection (MCODE) was used to identify the hub genes. Then the overlapping hub genes were predicted.

### Results

Two molecular subtypes were found in women with PCOS based on the expression similarity of preterm labor or labor-related genes, in which two modules were highlighted. The key

**Data Availability Statement:** The data that support the findings of this study are available in NCBI-GEO database of GEO accession at GSE84958 (https://www.ncbi.nlm.nih.gov/geo/query/acc.cgi?acc=

GSE84958), GSE5090 (https://www.ncbi.nlm.nih.gov/geo/query/acc.cgi?acc=GSE5090), GSE43264 (https://www.ncbi.nlm.nih.gov/geo/query/acc.cgi?acc=GSE43264) and GSE98421 (https://www.ncbi.nlm.nih.gov/geo/query/acc.cgi?acc=GSE98421).

**Funding:** This study was supported by the National Natural Science Foundation of China (81873837 to Fangfang Wang and 81973900 to Jue Zhou). The funders had no role in study design, data collection and analysis, decision to publish, or preparation of the manuscript.

**Competing interests:** The authors have declared that no competing interests exist.

modules and PPI network have five overlapping five hub genes, two of which, GTF2F2 and MYO6 gene, were further confirmed by the comparison between clustering subgroups according to the expression of hub genes.

## Conclusions

Distinct PCOS molecular subtypes were identified with preterm labor or labor-related genes, which might uncover the potential mechanism underlying heterogeneity of clinical pregnancy complications in women with PCOS.

## Introduction

Polycystic ovary syndrome (PCOS) is one of the most common endocrine disorders in women of reproductive age [1]. Its primary features are menstrual dysfunction, hyperandrogenism and polycystic ovary, but highly heterogeneous [2]. The current treatment strategies for the patients with PCOS are to reduce insulin resistance in order to reach a reduction of compensatory hyperinsulinemia, and to improve the metabolic and ovulatory features. For the overweight and obese PCOS patients, although physical activity and lifestyle change are the first steps to achieve weight loss, insulin-sensitizer drugs are the recommended first-line therapy, and many new insights have also been provided in the strategies for PCOS [3]. Myo-inositol and d-chiro-inositol have very specific physiological roles, however, they should be evaluated on the patients' conditions before the treatment and the effects of inositol therapy on different PCOS phenotypes needs further investigation [4]. Moreover, as PCOS causes a rising risk of maternal, fetal, and neonatal complications, including pregnancy-induced hypertension, preeclampsia, gestational diabetes mellitus, spontaneous preterm birth, an increased necessity for a cesarean section, elevated neonatal morbidity, prematurity, fetal growth restriction, birth weight variations, and transfer to the Neonatal Intensive Care Unit, a closer follow-up should be offered to PCOS women during pregnancy [5]. Although the causes of PCOS remain obscure, it is underpinned by a complex genetic and epigenetic architecture [1, 6, 7]. PCOS and PCOS-related gestational complications influence the intrauterine environment, leading to adverse developmental programming of the offspring for long-term, chronic health conditions [8, 9]. As preterm birth affects 1 in 10 pregnancies worldwide [10], the women with PCOS seemed to have a 2-fold increased risk of preterm labor, including both spontaneous preterm labor and indicated preterm labor which attributes to certain medical scenarios [11, 12].The preterm labor was associated with the substantial infant morbidity and mortality, long-term consequences of offspring as well as a huge socioeconomic cost [13–15]. Although the etiology of spontaneous preterm birth and the mechanism of labor is complex and unclear, a series of candidate genes have been reported to be involved in the preterm labor and labor [16–25].

In the past decade, the wide application of microarray technology and accurate RNA-sequencing technology has made it more convenient to reveal the mechanism underlying complex diseases (such as PCOS), on the basis of which our recent work uncovered gene biomarkers and developed a novel diagnostic model of PCOS [26]. Here, to elaborate the heterogeneity of preterm labor risk in women with PCOS, we analyzed the expression of previously reported preterm labor or labor related genes in PCOS based on public database- Gene Expression Omnibus (GEO) database, and attempted to classified PCOS into molecular subtypes through bioinformatics analysis.

**Table 1. Gene expression data from Gene Expression Omnibus (GEO) database.**

| Dataset ID | PCOS | Data type | Tissue type | Country |
|---|---|---|---|---|
| GSE84958 | 15 | RNA-seq | adipose | UK |
| GSE5090 | 9 | microarray | omental adipose | Spain |
| GSE43264 | 8 | microarray | Subcutaneous adipose | Ireland |
| GSE98421 | 4 | microarray | subcutaneous adipose | USA |

## Materials and methods

### Data sources

The NCBI-GEO database was searched for screening expression datasets, including microarray and RNA-seq, in women with PCOS. To minimize the heterogeneity among various tissues, four independent expression datasets in adipose tissue of women with PCOS were finally selected. The expression profiling by high throughput sequencing GSE84958, GSE5090, GSE43264 and GSE98421 were based on GPL16791, GPL96, GPL15362 and GPL570 platforms, with sample size of 15, 9, 8 and 4, respectively (Table 1).

### Collecting preterm or labor related genes

A literature review of English language studies was undertaken in the PubMed databases until November 23, 2020. Two independent review authors (JZ & ZJ) manually extracted the preterm or labor related genes from each eligible article, relevant review articles or book. Any disagreements were resolved by discussion with a third review author (FW).

### Data preprocessing

All datasets were downloaded as txt files, and outputs from mRNA array and RNA-seq were normal-exponential background corrected and then between-arrays quantile normalized using limma R package. Unsupervised cluster analysis of preterm or labor related genes was performed using the Consensus Cluster Plus R package (1.46.0) to select the best cluster group. Differential expression analysis of subgroups was performed using the Limma R package (3.36.5). The differential expressed genes were determined by two criteria: 1) the threshold value was greater than 1.0, and 2) the p-value calculated from pooled t-test was less than 0.05 and the corresponding confidence intervals were 95% [27].

### Identification of molecular subtypes of PCOS

The consensus k means clustering was utilized to perform consistent clustering and selecting of PCOS molecular subtypes based on preterm or labor related gene expression profiles. The optimal cluster number was determined by cumulative distribution function (CDF) curves of the consensus score, clear separation of the consensus matrix heatmaps, characteristics of the consensus cumulative distribution function plots, and adequate pair wise–consensus values between cluster members [26]. Principal Components Analysis (PCA) was used for confirmation of molecular clusters of PCOS samples with R package ggplot2.

### Functional annotation of the key module genes

We used weighted gene co-expression network analysis (WGCNA) R package to determine the genes correlated to molecular subtypes within all expressed genes in four GEO datasets. Then, Gene Ontology (GO) and KEGG (Kyoto Encyclopedia of Genes and Genomes) analysis

were performed to elaborate the functions and associated pathways of the key module genes in PCOS subtypes using ClusterProfiler R packages with P < 0.05 as the significance threshold.

## Construction of protein-protein interaction (PPI) networks

In order to determine the molecular mechanisms of signaling pathways and cellular activities in PCOS, the PPI network of the key module genes was constructed and visualized using the STRING (https://string-db.org) database.

## Prediction of hub genes in PCOS

Hub genes in the key modules were selected using CytoHubba through connection degree method. The Cytoscape software (http://www.cytoscape.org) was utilized to yeild the top 10 hub genes in PPI network using top degree method. Molecular Complex Detection (MCODE) was used to identify key clusters of genes within PPI network. Finally, we summarized the overlapping genes between results of MCODE and CytoHubba to create a consensus of predictions to identify more accurate hub genes [26].

# Results

## Characteristics of datasets and patients

After search strategy and selection, four mRNA datasets, which was from UK, Spain, Ireland and USA, were finally enrolled in current study with the total sample size of 36. One dataset GSE84958 was got from RNA-seq analysis, and all the rest datasets was got from Array analysis. Since the women in GSE5090 dataset were diagnosed as PCOS with the presence of oligoovulation, clinical and/or biochemical hyperandrogenism in 2006 [28], whereas the diagnostic criteria of the other studies in 2014, 2017 and 2018 were not available. The PCOS patients in GSE5090 underwent bariatric surgery because of morbid obesity, while those in GSE98421 were lean.

## The genes related to preterm labor or labor

According to previously published literature [16–25] and unpublished data of our group, genes related to preterm labor or labor were yielded, and shown in Fig 1A. Four genes played roles in induction of uterine contraction, including *hematopoietic prostaglandin D synthase (HPGDS)*, *aldo-keto reductase family 1 member C3 (AKR1C3) and ATP binding cassette subfamily C member 4 (ABCC4)* as well as *corticotropin releasing hormone (CRH) and its receptor (CRHR1)*. Seven genes were associated with inflammation and immune response, including *interleukin 6 (IL6), tumor necrosis factor (TNF), interleukin 1 beta (IL1B), complement C3 (C3), complement factor H (CFH), complement C1r (C1R), toll like receptor 8 (TLR8)* and *endoplasmic reticulum aminopeptidase 2 (ERAP2)*. Three genes were suggested as transcription regulators, including *sirtuin 1 (SIRT1), tripartite motif containing 28 (TRIM28), nuclear factor kappa B subunit 1 (NFKB1)*. There existed seven genes which influenced cell proliferation, migration, adhesion and metabolism: *ADAM metallopeptidase with thrombospondin type 1 motif 12 (ADAMTS12), ADAMTS16, insulin like growth factor binding protein 1 (IGFBP1), IGFBP2, IGFBP6, tenascin C (TNC), Fos proto-oncogene (FOS) and FosB proto-oncogene (FOSB)*. Proteins coded by *hydroxysteroid 17-beta dehydrogenase 4 (HSD17B4), androgen receptor (AR), estrogen receptor 1 (ESR1), peroxisome proliferator activated receptor gamma (PPARG)* gene-functioned in metabolism and function of steroids. Next, a correlation analysis was performed to explore the correlation among the genes of interest (Fig 1B).

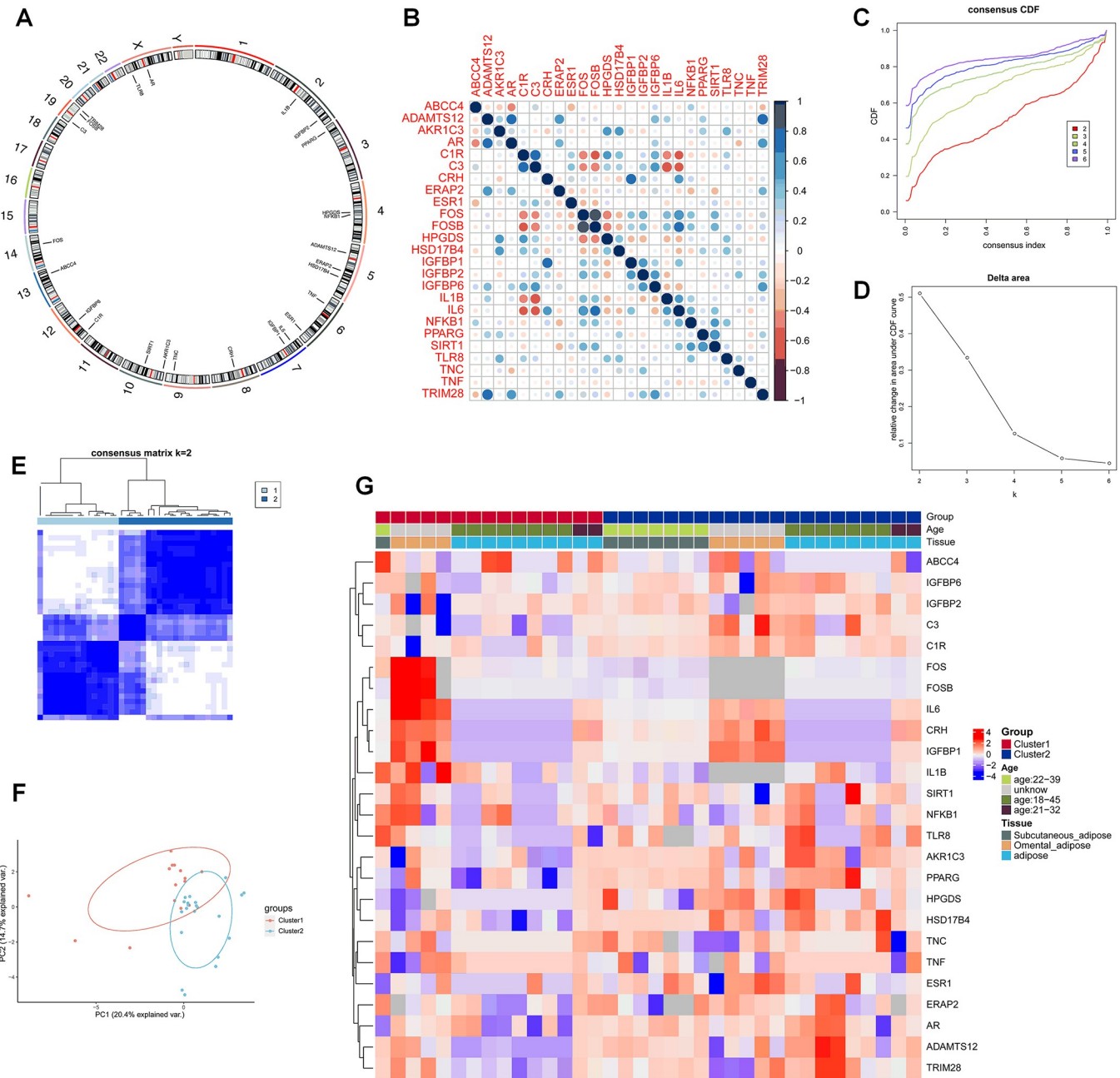

**Fig 1. Identification of molecular subtypes in PCOS based on preterm labor or labor-related genes.** (A) Chromosomal distribution diagram of preterm labor or labor-related genes involved in the present study. (B) The relationship among preterm labor or labor-related genes of interest were shown. (C) The cumulative distribution function (CDF) curves of consensus scores based on different subtype number (k = 2, 3, 4, 5, 6) and the corresponding color are represented. (D) The CDF Delta area curve of all samples when k = 2. (E) A relative stable partitioning of the samples at k = 2 in consensus heatmap. (F) In PCA analysis, the symbols represent the gene expressed differently in two clusters. (G) The expression heatmap of the preterm labor or labor-related genes among various molecular and clinical subtypes.

## The molecular subtyping in PCOS based on preterm labor or labor-related genes

Based on the expression similarity of preterm labor or labor-related genes, women with PCOS were divided into two molecular subtypes with clustering stability k = 2 (Fig 1C–1E). The

clustering classification of two subgroups in patients with PCOS was verified with principal component analysis (PCA) (Fig 1F). Fig 1G indicated a distinct expression pattern in the genes of interest profiles between the two molecular subtypes, and age group from 22–39 and subcutaneous distribution of adipose tissue was mainly included in cluster 2.

## Heterogeneity of biological process in key modules of PCOS

WGCNA identified 4 modules in the PCOS population (Fig 2A–2C). Cluster 1 negatively correlated with 4 modules, whereas cluster 2 positively correlated with 4 modules. Number of the related genes in each module was as follow: 4, 36, 28 and 80 genes for brown, blue, grey and turquoise modules, respectively. Thus, we used blue and turquoise modules related genes to furthermore explore biological function.

GO enrichment and KEGG pathway analyses were conducted. With GO analyses, genes in the blue and turquoise modules were found to be primarily associated with phospholipid metabolism process, especially phosphotransferase activity, as well as DNA transcription (Fig 2D–2F). Additionally, KEGG pathway analysis lent support to the above result, and phospholipid metabolism and DNA transcription related pathways were enriched (Fig 2G).

## Hub genes for PCOS

On one hand, to screen the upstream regulators with high connectivity, we identified the 81 hub genes for the key modules (blue and turquoise); On the other hand, following PPI networks construction, top 10 hub genes were identified with Cytoscape (Fig 3A). Finally, five hub genes were found overlapped between the above two analyses and they were considered as hub genes for PCOS, including MYO6, ACTL6A, NCBP2, GTF2F2 and MRPL13 (Fig 3B).

## Differential expression of hub genes between subtypes in PCOS

To confirm the roles of hub genes in PCOS subtypes, we compared the expression of the five hub genes mentioned above between various subtypes. First, we observed the significantly increased expression of all the five hub genes between cluster1 and cluster2 divided according to the expression of genes related to preterm labor or labor (Fig 3C). Then, we investigated the expression of these hub genes among adipose tissue subtypes (S1A Fig) and various age subgroups (S1B Fig), but did not find any difference. To understand the expression correlation among the five hub genes, we carried out a Pearson analysis, and found that they were positively related with each other (Fig 3D).

## Grouping by cluster analysis based on hub genes of PCOS

Based on the five hub genes, cluster analysis revealed that the 36 women with PCOS could be classified into 2 subgroups: hub-cluster1 and hub-cluster2 (Fig 3E and 3F). The expression patterns of the five hub genes in both clinical subgroups were showed as heatmap (Fig 3G). To further confirm the impacts of hub genes, the expression comparisons were performed between the clinical subgroups, and then a significant change was found for GTF2F2 and MYO6 gene (Fig 3H). What's more, we made a Sankey diagram to visualize the links among preterm labor or labor related gene clusters, hub gene clusters, age subgroups and adipose tissue subtypes (Fig 3I).

## Discussion

The development of machine learning algorithms and the availability of gene expression data in the public databases provide approaches to infer biomarkers for disease diagnosis or

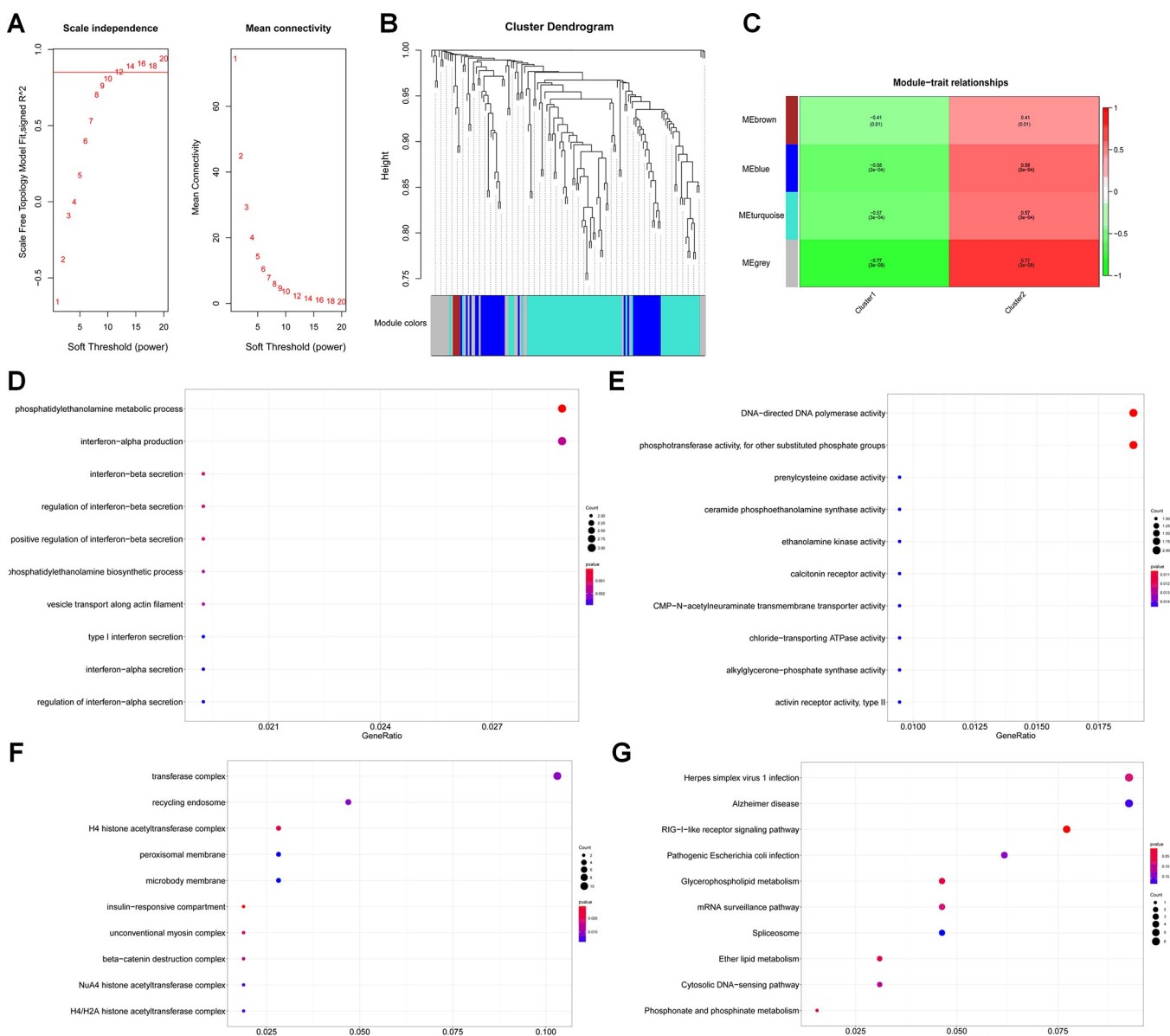

**Fig 2. Construction of expression modules by WGCNA package.** (A) Analysis of the scale-free fit index for various soft-thresholding powers and analysis of the mean connectivity for various soft-thresholding powers. (B) The cluster dendrogram of genes. Each branch represents one gene, and every color below represents one expression module. (C) Heatmap of the correlation between module genes and the clusters. (D-F) Top 10 terms from a GO analysis of molecular function, biological process and cellular component in the blue and turquoise modules. (G) Top 10 terms were clustered by KEGG pathway analysis in the blue and turquoise modules.

prognosis in a wide range of fields [28–32]. The bioinformatic attempts for PCOS vary from susceptibility and pathogenesis, to precise diagnosis and tailed therapy [33–37]. According to the most widely used Rotterdam PCOS diagnostic criteria for adult, any 2 out of 3 following features should be met: androgen excess, ovulatory dysfunction, and polycystic ovaries, suggesting that the clinical manifestations and pregnancy complications of PCOS are highly heterogeneous [38–42]. Thus, it is important to find a way to differentiate the heterogeneity of preterm labor risk, and then guide the clinical intervention. In the clinical practice, a feasible genetic test can be expected to perform for the PCOS patients with higher risks, however the

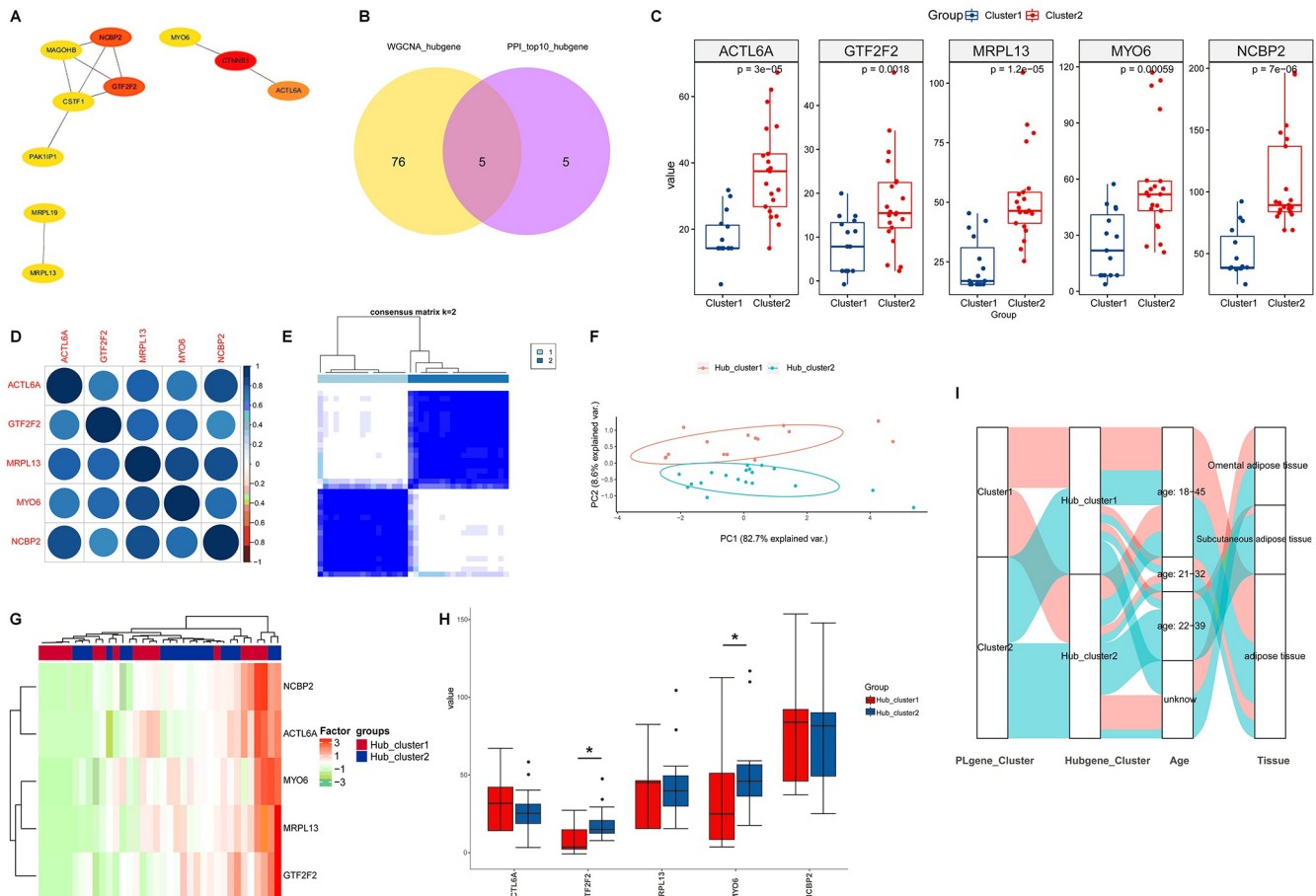

**Fig 3. Grouping based on hub genes of PCOS.** (A) Top10 hub genes were identified and PPI networks were established with Cytoscape. (B) Venn diagram for the overlapping genes between the above 10 hub genes and hub genes for the key modules. (C) The expression comparison of the five overlapped hub genes between cluster1 and cluster2. (D) The relationship among these overlapped hub genes were shown. (E) A relative stable partitioning of the samples at k = 2 in consensus heatmap. (F) In PCA analysis, the symbols represent the gene expressed differently in hub_cluster1 and hub_cluster2. (G) The expression heatmap of these overlapped hub genes in hub_cluster1 and hub_cluster2. (H) The expression comparison of the five overlapped hub genes between hub_cluster1 and hub_cluster2. (I) Sankey diagram for the links among the preterm labor or labor related gene clusters, hub gene clusters, age subgroups and adipose tissue subtypes of PCOS.

economic cost may be considered. Since there is conflicting evidence as to whether or not PCOS women predispose to preterm birth [43], it is reasonable for obstetricians to give the primary prevention strategy firstly, and then the secondary prevention strategies if necessary, to stratify subgroup of PCOS patients with genetic predisposition. The current research indicates that two molecular subtypes were identified in PCOS, by clustering based on the expression of candidate genes related preterm labor and labor. These two subtypes exhibited distinct biological processes and pathways. In addition, two hub genes were spotlighted to imply the key network nodes in the molecular subtypes of PCOS concerning to preterm labor. To our knowledge, this is the first study concerning the transcriptome-wide molecular subtyping of PCOS with preterm labor or labor associated genes.

There exist some possible explanations for the PCOS-related preterm labor. On one hand, hyperandrogenism, one of main PCOS features, usually gets enhanced throughout the pregnancy period, which might increase the risk of pregnancy complication, such as preterm labor [11]. Androgens could induce indicated preterm labor in PCOS patients due to severe

pregnancy complications, e.g. pre-eclampsia, possibly through changes of endovascular tro-phoblast invasion and placentation [44]. Androgens might increase the incidence of spontane-ous preterm labor in women with PCOS by acting on cervical remodelling and myometrial function [45]. On the other hand, there may exist other molecular mechanisms underlying the preterm labor risk in non-hyperandrogenic PCOS patients. For instance, an abnormal pattern of low-grade chronic inflammation in combination with a subclinical impairment of vascular structure and function were found in both non-pregnant and pregnant women with PCOS, probably contributing to the subsequent reduced depth of endovascular trophoblast and abnormal placentation [46]. As one of the many pro-inflammatory cytokines involved in the induction of spontaneous preterm labor, IL-6 stands out for its pleiotropic effects in both acute and chronic inflammation [47]. Our previous study indeed observed elevated IL-6 levels in peripheral blood of non-hyperandrogenic pregnant women with PCOS [48], suggesting a potential link of preterm labor to non-hyperandrogenic PCOS. The results suggested that genetics would be used to stratify a proportion of women with PCOS into the subgroups with clinical significance. The primary prevention method for the PCOS patients with genetic pre-disposition is to control the risk factors (e.g. obesity) through lifestyle modification during pre-pregnancy and early pregnancy. And the secondary prevention strategy for this PCOS sub-group is to apply appropriate cervical length surveillance, and the precise vaginal administra-tion of progesterone [43, 49]. Two hub genes of molecular subtyping in PCOS, GTF2F2 and MYO6, were highlighted in the current study. As GTF2F2 gene encodes general transcription factor IIF (TFIIF) subunit 2, the interaction between TFIIF and RPB5-mediating protein is critical to suppress the activated transcription [50], which might be involved in the biological events of energy metabolism, metabolic disorders and fertility [51–53]. MYO6 gene encodes a reverse-direction motor protein that moves toward the minus end of actin filaments and plays a role in intracellular vesicle and organelle transport [54, 55], and execute its functions at mul-tiple steps in autophagy, microtubule polymerization, cell proliferation and metastasis, and hearing loss, spermatogenesis [54, 56–59].

There are several limitations in the present study. First, our sample size was relatively small, and the ethnicities were restricted to Europe and North America. Second, we were not able to access more clinical features (e.g. gestational outcomes) and demographic data of the available samples selected in the present study for more detailed investigation. In addition, our analysis strategy is promising to facilitate the risk assessment for the preterm labor in PCOS, however, the further confirmation in diverse tissue types through biomedical experiments should be conducted sequentially due to the limitations of bioinformatics.

## Supporting information

**S1 Fig. The comparison of the five overlapped hub genes among various clinical subtypes of women with PCOS.** The expression of these hub genes was shown among adipose tissue subtypes (A) and various age subgroups (B).
(TIF)

## Author Contributions

**Conceptualization:** Jue Zhou, Songying Zhang, Fangfang Wang.

**Data curation:** Jue Zhou, Zhou Jiang, Fangfang Wang.

**Formal analysis:** Jue Zhou, Zhou Jiang, Leyi Fu, Fan Qu, Minchen Dai, Ningning Xie.

**Investigation:** Jue Zhou, Zhou Jiang, Fangfang Wang.

**Methodology:** Jue Zhou, Songying Zhang, Fangfang Wang.

**Project administration:** Songying Zhang, Fangfang Wang.

**Supervision:** Songying Zhang, Fangfang Wang.

**Visualization:** Jue Zhou, Fangfang Wang.

**Writing – original draft:** Jue Zhou, Fangfang Wang.

**Writing – review & editing:** Jue Zhou, Fangfang Wang.

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
