## [Decision Letter · Decision Letter 0]

24 Nov 2022

PONE-D-22-28834Contribution of labor related gene subtype classification on heterogeneity of polycystic ovary syndromePLOS ONE

Dear Dr. Wang,

Thank you for submitting your manuscript to PLOS ONE. After careful consideration, we feel that it has merit but does not fully meet PLOS ONE’s publication criteria as it currently stands. Therefore, we invite you to submit a revised version of the manuscript that addresses the points raised during the review process.

We look forward to receiving your revised manuscript.

Kind regards,

Antonio Simone Laganà, M.D., Ph.D.

Academic Editor

PLOS ONE

Journal Requirements:

**Additional Editor Comments:**

The topic of the manuscript is interesting. Nevertheless, the reviewers raised several concerns: considering this point, I invite authors to perform the required major revisions.

Reviewers' comments:

Reviewer's Responses to Questions

**Comments to the Author**

1. Is the manuscript technically sound, and do the data support the conclusions?

Reviewer #1: Yes

Reviewer #2: Yes

2. Has the statistical analysis been performed appropriately and rigorously? 

Reviewer #1: No

Reviewer #2: Yes

3. Have the authors made all data underlying the findings in their manuscript fully available?

Reviewer #1: Yes

Reviewer #2: Yes

4. Is the manuscript presented in an intelligible fashion and written in standard English?

Reviewer #1: No

Reviewer #2: Yes

5. Review Comments to the Author

Reviewer #1: I read with great interest the Manuscript titled “Contribution of labor related gene subtype classification on heterogeneity of polycystic ovary syndrome” (PONE-D-22-28834), which falls within the aim of this Journal.

In my honest opinion, the topic is interesting enough to attract the readers’ attention. Methodology is accurate and conclusions are supported by the data analysis.

Nevertheless, authors should clarify some point and improve the discussion citing relevant and novel key articles about the topic.

Authors should consider the following recommendations:

- Manuscript should be further revised by a native English speaker.

- In the Results section, the Authors have simply reported the p values, from which however it is not possible to deduce the real clinical relevance of the highlighted statistical significance. In order to better understand the obtained results, I suggest reporting not only the p values, but also the corresponding confidence intervals.

- Does this manuscript conform the Enhancing the QUAlity and Transparency Of health Research (EQUATOR) network guidelines? It would be mandatory to declare about this element.

- I recommend to highlight, at least briefly, current management strategies for PCOS (PMID: 35472446; PMID: 3407070) and reproductive/obstetric outcomes in this population (PMID: 33876903).

Reviewer #2: The manuscript is well written and complete.

My suggestions are:

- How your results can affect the clinical practice? Is a genetic test feasible and useful in these patients? Is it economically sustainable?

- If a patient have a genetic predisposition, what is the strategy to reduce risk of preterm birth?

Results are very interesting but very limited if they don't improve outcome. Please, you have to explain the primary and secondary prevention methods that can be applied in these patients.

MINOR REVISIONS

- Page 5, lines 96,97: please correct “preterm” labour and add space between “genes” and “was”

- Page 8, line 158: please correct “genes” not gene

- Page 12, line 250: please correct “genes” not gens

6. PLOS authors have the option to publish the peer review history of their article (what does this mean?). If published, this will include your full peer review and any attached files.

Reviewer #1: **Yes: **Alessandra Lopez

Reviewer #2: **Yes: **Basilio Pecorino

---

## [Author Response · Author response to Decision Letter 0]

2 Feb 2023

RE: PLOS ONE - Decision on No.PONE-D-22-28834

Title: Contribution of labor related gene subtype classification on heterogeneity of polycystic ovary syndrome

Dear Dr. Laganà:

Thank you for inviting a revision of our manuscript, which we are now submitting for consideration for publication. 

We would like to thank you for the constructive and positive comments. We have revised the manuscript in response to the suggestions of reviewers where appropriate. The corrections and amendments have been highlighted RED in the revised manuscript. We also provided point-by-point replies to each concern raised by the reviewers and editor as listed below. The revised manuscript has been uploaded in the submission system of the journal.

We believe that our revised manuscript is greatly improved, and we hope that it is now more acceptable for publication in PLOS ONE.

Thank you. 

With best wishes!

Sincerely,

Fang-Fang Wang

Replies to the comments

Journal Requirements:

Response: Thank you for the comment! As suggested, we have ensured that the revised manuscript meets PLOS ONE's style requirements.

Response: Thank you for the comment! As suggested, we have ensured that we have provided the correct grant numbers in the “Funding Information” section in the submission system. 

This study was supported by the National Natural Science Foundation of China (81873837 to Fangfang Wang and 81973900 to Jue Zhou). The funders had no role in study design, data collection and analysis, decision to publish, or preparation of the manuscript. The funding information was provided in the file “Funding statement” which has also been uploaded in the submission system.

Response: Thank you for the comment! As suggested, we have added the URLs of the resource of RNA-seq and microarray in the part of data availability statement of the revised manuscript.

Additional Editor Comments:

The topic of the manuscript is interesting. Nevertheless, the reviewers raised several concerns: considering this point, I invite authors to perform the required major revisions.

Reviewers' comments:

Reviewer's Responses to Questions

Comments to the Author

1. Is the manuscript technically sound, and do the data support the conclusions?

Reviewer #1: Yes

Reviewer #2: Yes

2. Has the statistical analysis been performed appropriately and rigorously?

Reviewer #1: No

Reviewer #2: Yes

3. Have the authors made all data underlying the findings in their manuscript fully available?

Reviewer #1: Yes

Reviewer #2: Yes

4. Is the manuscript presented in an intelligible fashion and written in standard English?

Reviewer #1: No

Reviewer #2: Yes

5. Review Comments to the Author

Reviewer #1: I read with great interest the Manuscript titled “Contribution of labor related gene subtype classification on heterogeneity of polycystic ovary syndrome” (PONE-D-22-28834), which falls within the aim of this Journal.

In my honest opinion, the topic is interesting enough to attract the readers’ attention. Methodology is accurate and conclusions are supported by the data analysis.

Nevertheless, authors should clarify some point and improve the discussion citing relevant and novel key articles about the topic.

Authors should consider the following recommendations:

1. Manuscript should be further revised by a native English speaker.

Response: Thank you for the comment! As suggested, we have invited a native English speaker to improve the language of the present manuscript.

2 - In the Results section, the Authors have simply reported the p values, from which however it is not possible to deduce the real clinical relevance of the highlighted statistical significance. In order to better understand the obtained results, I suggest reporting not only the p values, but also the corresponding confidence intervals.

Response: Thank you for the comment! As suggested, we have mentioned the p-value was set to less than 0.05, and the corresponding confidence intervals were 95% in the revised manuscript.

3- Does this manuscript conform the Enhancing the QUAlity and Transparency Of health Research (EQUATOR) network guidelines? It would be mandatory to declare about this element.

Response: Thank you for the comment! As there is no available study type in the reporting guidelines for the present study, we did not declare about this element in the original version, however, we have fully considered these guidelines when reporting the present study. 

4- I recommend to highlight, at least briefly, current management strategies for PCOS (PMID: 35472446; PMID: 3407070) and reproductive/obstetric outcomes in this population (PMID: 33876903).

Response: Thank you for the comment! Yes, we fully agree with this. As suggested, we have highlighted the current management strategies for PCOS and the reproductive/obstetric outcomes in this population in the revised manuscript. We also added the three papers (PMID: 35472446; PMID: 3407070; PMID: 33876903) as new references in the revised manuscript. 

Reviewer #2: The manuscript is well written and complete.

My suggestions are:

1. - How your results can affect the clinical practice? Is a genetic test feasible and useful in these patients? Is it economically sustainable?

Response: Thank you for the comment! As suggested, we have added the contents in the discussion part of the revised manuscript. 

2.- If a patient have a genetic predisposition, what is the strategy to reduce risk of preterm birth?

Response: Thank you for the comment! As suggested, we have added the contents in the discussion part of the revised manuscript. 

3. Results are very interesting but very limited if they don't improve outcome. Please, you have to explain the primary and secondary prevention methods that can be applied in these patients.

Response: Thank you for the comment! As suggested, we have added the contents in the discussion part of the revised manuscript. 

MINOR REVISIONS

- Page 5, lines 96,97: please correct “preterm” labour and add space between “genes” and “was”

- Page 8, line 158: please correct “genes” not gene

- Page 12, line 250: please correct “genes” not gens

Response: Thank you for the comment! As suggested, we have revised all of the contents above in the revised manuscript. 

6. PLOS authors have the option to publish the peer review history of their article (what does this mean?). If published, this will include your full peer review and any attached files.

Do you want your identity to be public for this peer review? For information about this choice, including consent withdrawal, please see our Privacy Policy.

Reviewer #1: Yes: Alessandra Lopez

Reviewer #2: Yes: Basilio Pecorino

---

## [Decision Letter · Decision Letter 1]

14 Feb 2023

Contribution of labor related gene subtype classification on heterogeneity of polycystic ovary syndrome

PONE-D-22-28834R1

Dear Dr. Wang,

We’re pleased to inform you that your manuscript has been judged scientifically suitable for publication and will be formally accepted for publication once it meets all outstanding technical requirements.

Kind regards,

Antonio Simone Laganà, M.D., Ph.D.

Academic Editor

PLOS ONE

Additional Editor Comments (optional):

Authors performed the required corrections, which were positively evaluated by the reviewers. I am pleased to accept this paper for publication.

Reviewers' comments:

Reviewer's Responses to Questions

**Comments to the Author**

1. If the authors have adequately addressed your comments raised in a previous round of review and you feel that this manuscript is now acceptable for publication, you may indicate that here to bypass the “Comments to the Author” section, enter your conflict of interest statement in the “Confidential to Editor” section, and submit your "Accept" recommendation.

Reviewer #1: (No Response)

Reviewer #2: All comments have been addressed

2. Is the manuscript technically sound, and do the data support the conclusions?

Reviewer #1: (No Response)

Reviewer #2: Yes

3. Has the statistical analysis been performed appropriately and rigorously? 

Reviewer #1: (No Response)

Reviewer #2: Yes

4. Have the authors made all data underlying the findings in their manuscript fully available?

Reviewer #1: (No Response)

Reviewer #2: Yes

5. Is the manuscript presented in an intelligible fashion and written in standard English?

Reviewer #1: (No Response)

Reviewer #2: Yes

6. Review Comments to the Author

Reviewer #1: (No Response)

Reviewer #2: Dear authors, all comments by reviewers have be addressed. Thank you very much for revisions submitted

7. PLOS authors have the option to publish the peer review history of their article (what does this mean?). If published, this will include your full peer review and any attached files.

Reviewer #1: **Yes: **Alessandra Lopez

Reviewer #2: **Yes: **BASILIO PECORINO

---

## [Editor Report · Acceptance letter]

20 Feb 2023

PONE-D-22-28834R1 

Contribution of labor related gene subtype classification on heterogeneity of polycystic ovary syndrome 

Dear Dr. Wang:

I'm pleased to inform you that your manuscript has been deemed suitable for publication in PLOS ONE. Congratulations! Your manuscript is now with our production department. 

Kind regards, 

on behalf of

Dr. Antonio Simone Laganà 

Academic Editor

PLOS ONE